# Genome-Wide Association Study of Phenylalanine Derived Glucosinolates in *Brassica rapa*

**DOI:** 10.3390/plants11091274

**Published:** 2022-05-09

**Authors:** Guoxia Shang, Huiyan Zhao, Linhui Tong, Nengwen Yin, Ran Hu, Haiyan Jiang, Farah Kamal, Zhi Zhao, Liang Xu, Kun Lu, Jiana Li, Cunmin Qu, Dezhi Du

**Affiliations:** 1Laboratory for Research and Utilization of Qinghai Tibet Plateau Germplasm Resources, Key Laboratory of Spring Rape Genetic Improvement, Academy of Agricultural and Forestry Sciences, Qinghai University, Xining 810016, China; shangguoxia@126.com (G.S.); zhaozhi918@sohu.com (Z.Z.); qhrapelab@126.com (L.X.); 2Chongqing Engineering Research Center for Rapeseed, College of Agronomy and Biotechnology, Southwest University, Chongqing 400716, China; zhaohuiyan@swu.edu.cn (H.Z.); t15565867152@email.swu.edu.cn (L.T.); nwyin80@swu.edu.cu (N.Y.); hr1996@email.swu.edu.cn (R.H.); jhy20191712@email.swu.edu.cn (H.J.); farah.kamal93@outlook.com (F.K.); drlukun@swu.edu.cn (K.L.); 3Academy of Agricultural Sciences, Southwest University, Chongqing 400715, China; 4Engineering Research Center of South Upland Agriculture, Ministry of Education, Chongqing 400715, China

**Keywords:** benzylglucosinolate, phenethylglucosinolate, 2-phenylethylglucosinolate, glucosinolate biosynthesis and regulation, *Brassica rapa*, genome-wide association study, LC-MS

## Abstract

Glucosinolates (GSLs) are sulfur-containing bioactive compounds usually present in Brassicaceae plants and are usually responsible for a pungent flavor and reduction of the nutritional values of seeds. Therefore, breeding rapeseed varieties with low GSL levels is an important breeding objective. Most GSLs in *Brassica rapa* are derived from methionine or tryptophan, but two are derived from phenylalanine, one directly (benzylGSL) and one after a round of chain elongation (phenethylGSL). In the present study, two phenylalanine (Phe)-derived GSLs (benzylGSL and phenethylGSL) were identified and quantified in seeds by liquid chromatography and mass spectrometry (LC-MS) analysis. Levels of benzylGSL were low but differed among investigated low and high GSL genotypes. Levels of phenethylGSL (also known as 2-phenylethylGSL) were high but did not differ among GSL genotypes. Subsequently, a genome-wide association study (GWAS) was conducted using 159 *B. rapa* accessions to demarcate candidate regions underlying 43 and 59 QTNs associated with benzylGSL and phenethylGSL that were distributed on 10 chromosomes and 9 scaffolds, explaining 0.56% to 70.86% of phenotypic variations, respectively. Furthermore, we find that 15 and 18 known or novel candidate genes were identified for the biosynthesis of benzylGSL and phenethylGSL, including known regulators of GSL biosynthesis, such as *BrMYB34*, *BrMYB51*, *BrMYB28*, *BrMYB29* and *BrMYB122*, and novel regulators or structural genes, such as *BrMYB44*/*BrMYB77* and *BrMYB*60 for benzylGSL and *BrCYP79B2* for phenethylGSL. Finally, we investigate the expression profiles of the biosynthetic genes for two Phe-derived GSLs by transcriptomic analysis. Our findings provide new insight into the complex machinery of Phe-derived GSLs in seeds of *B. rapa* and help to improve the quality of Brassicaceae plant breeding.

## 1. Introduction

The glucosinolates (GSLs) constitute a group of sulfur-containing and amino acid-derived secondary metabolites, which play important roles in plant defense and human health and are present in all plants of the Brassicaceae family [1,2]. Rapeseed is one of the major oil crops grown worldwide, including *Brassica rapa* (AA, 2n = 20), *Brassica napus* (AACC, 2n = 38) and *Brassica juncea* (AABB, 2n = 36). After oil extraction of the seeds, the GSLs remaining in pressed rapeseed meal will significantly reduce its nutritional value [3,4]. Therefore, breeding rapeseed with low content of GSLs in seeds is an indispensable objective.

The GSLs are derived from amino acid-derived oximes that are subsequently decorated with a thioglucose moiety and a sulfate group. Based on the precursor amino acid and the types of elongation and modification to the side chain, GSLs can be classified into three classes, including aliphatic GSLs derived from methionine (Met), valine (Val), leucine (Leu) and isoleucine (Ile), indolic GSLs derived from tryptophane (Trp) and benzenic GSLs derived from phenylalanine (Phe) and tyrosine (Tyr) [5,6]. In *B. rapa*, the main GSLs are derived from Met, Trp and Phe [7]. Another classification is based on chain elongation or not. Some GSLs are biosynthesized directly from usual protein amino acids, this group includes the Trp-derived indolylmethylGSL and benzylGSL derived from Phe. The steps from amino acid to GSL are known as the core structure biosynthesis. However, the biosynthesis of most GSLs involves an initial chain elongation of the precursor amino acid. This group of GSLs include Phe-derived phenethylGSL (2-Phenylethyl GSL) and the Met-derived GSLs. The biosynthesis of phenethylGSL involves a single round of chain elongation, resulting in one extra CH_2_ group in the chemical structure, while the biosynthesis of Met-derived GSLs can involve several rounds of chain elongation. The group of chain-elongated GSLs begin their biosynthesis with removal of the amino group by a branched-chain amino acid aminotransferase (BCAT4), followed by import to the chloroplast by bile acid transporter (BAT5), where the chain elongation takes place. As we know, the chain elongation includes a three-step transformation: condensation with acetyl-CoA by methylthioalkylmalate synthase enzymes (MAM), isomerization by an isopropylmalate isomerase (IPMI, IIL) and oxidative decarboxylation by an isopropylmalate dehydrogenase (IMDH), yield chain-elongated 2-oxo acid. More cycles of elongation processes would happen for Met derivates, as for phenethylGSL, the 2-oxo acid with one more methylene group is exported to the cytosol and finally transamination by BCAT3 to enter the core structure biosynthesis mentioned above [5,6,8,9]. Overall, the difference between biosynthesis of phenethylGSL and benzylGSL includes both presence or absence of chain elongation as well as different CYP79 enzymes in the first step of the core structure biosynthesis [6,10,11,12,13]. Many structural genes are shared among the biosynthesis of phenethylGSL and the Met-derived GSLs [1,10], and it is uncertain what factors control the relative flux to Met-derived GSLs and phenethylGSL.

In plants, the GSL pathway has also been well known as a ‘model’ for secondary metabolites [10]. To date, several MYB transcription factors have been verified to regulate GSL biosynthesis in different plants. For example, higher expression of *BoMYB28* increased the content of Met-derived 4-(methylsulfinyl) butylGSL in broccoli (*Brassica oleracea* var *italica*), and targeted silencing of *BjMYB28* homologs in *Brassica juncea* significantly reduced GSLs derived from Met in seeds [14,15]; overexpressing *BoMYB29* homologs could upregulate GSL biosynthesis from Met in *B. oleracea* [16], and *BrMYB34*, *BrMYB51* and *BrMYB122* act together to control biosynthesis of Trp-derived GSLs in *A. thaliana* [17]. Taken together, there is consensus that *MYB28*, *MYB29* and *MYB76* regulate the biosynthesis of Met-derived GSLs, while the biosynthesis of Trp-derived GSLs is regulated by *MYB34*, *MYB51* and *MYB122*. In contrast to the rather well understood regulation of GSL biosynthesis from Met and Trp, the regulation of the biosynthesis of both benzylGSL (without chain elongation) and phenethylGSL (with chain elongation) is poorly understood.

Genome-wide association studies (GWAS) associate phenotypes with genes on a genomic scale. It is verified that metabolite-based GWAS (mGWAS) is a powerful method for the screening of candidate genes for different metabolites and dissect complex traits in plants [18,19]. Moreover, liquid chromatography and mass spectrometry (LC-MS) analysis has been widely used for qualitative and quantitative analysis on GSLs in different plant tissues [20,21,22]. To dissect the genetic basis of two Phe-derived GSLs in *B. rapa,* we intend to detect QTNs (quantitative trait nucleotides, which link with the tested quantitative trait and are detected by multi-locus random-SNP-effect mixed linear model tools (mrMLM)), candidate genes and regulatory networks associated with benzylGSL and phenethylGSL of *B. rapa* by LC-MS, GWAS and transcriptome analysis. In this study, numerous QTNs, significant genes and characterized biosynthesis pathways were potentially involved in the regulation of benzylGSL and phenethylGSL in *B. rapa*. The results will help us to better understand genetic basis of two Phe-derived GSLs in *B. rapa* and lay a foundation for improving the quality of rapeseed.

## 2. Results

### 2.1. The Dynamic Accumulation of Phe-Derived Glucosinolates in B. rapa Seeds

We studied seed development in GSL levels as a function of days after pollination (DAP). The qualitative and quantitative analysis of benzylGSL (C91) and phenethylGSL (C121) were performed in 15 DAP, 25 DAP, 35 DAP, 45 DAP and 50 DAP seeds of BrHG (*B. rapa* line with high content of total GSLs in seeds) and BrLG (*B. rapa* line with low content of total GSLs in seeds). BenzylGSL and phenethylGSL were identified in LC-MS data from retention time, m/z of the molecular ion [M-H](-) and the occurrence of the common GSL fragment HSO4(-) at m/z 97 (Figure 1). The retention time of phenethylGSL (C121 at 6.16 min) was higher than for benzylGSL (C91 at 4.31 min) in agreement with the chemical structures. The content of phenethylGSL in seeds was much higher than benzylGSL. The highest level of benzylGSL existed in 20 DAP seeds of BrLG and declined in pace with seed maturation, on the contrary, the content of benzylGSL increased with seed maturity in BrHG and was with highest level in 50 DAP seeds. No accumulation of phenethylGSL was detected in young seeds (15 DAP and 25 DAP seeds) of *B. rapa* (Figure 1c,d) The accumulation trends of phenethylGSL content were similar between BrHG and BrLG, which both increased with seed maturity.

### 2.2. QTNs (Quantitative Trait Nucleotides) Detected for Two Phe-Derived Glucosinolates

In this study, 43 and 59 QTNs were detected for benzylGSL (C91) and phenethylGSL (C121) respectively, and distributed on 10 chromosomes and 9 scaffolds (Appendix A). Eleven and 10 QTNs were repeatedly detected by two or three different GWAS algorithms for benzylGSL and phenethylGSL on different chromosomes, but the QTNs detected for the two Phe-derived GSLs were all different from each other (Figure 2, Appendix A). Importantly, the contribution rate of QTNs ranged from 0.56% to 70.86%, and 21 QTNs of them were over 10% (9 QTNs for benzylGSL and 12 QTNs for phenethylGSL). It was noticed that one QTN with a high contribution rate for benzylGSL was strikingly detected on A2 chromosome (27.03 Mb), and two other QTNs with high contribution rate for phenethylGSL were detected on A9 (21.77 Mb) and A3 (32.77 Mb) chromosomes (Figure 2). The distribution of detected QTNs on chromosomes was uneven, and it was noticed that QTNs were gathered on different chromosomes. The QTN number in 1Mb of chromosome were calculated, and 5 hot regions with higher density of QTNs were obtained, which were distributed on A01 (8.15–9.25 Mb), A02 (10.8–12.7 Mb), A02 (26.05–27.50 Mb), A3 (21.0–22.95 Mb), and A8 (16.6–18.05 Mb), respectively. It was assumed that important regulators might be detected from five hot regions.

### 2.3. Screening of Candidate Genes for Two Phe-Derived Glucosinolates

According to the *B. rapa* reference genome (http://39.100.233.196/#/Download/, accessed on 1 June 2021), a total of 3218 genes were obtained near the detected QTNs for the two Phe-derived GSLs. Combined with the expression value of collected genes by RNA-seq, 748 DEGs were screened out from 3218 genes and were used as search terms for protein-protein interaction prediction with known GSL biosynthesis genes (Appendix A). Six genes interacting with known GSL biosynthesis genes were obtained, including *BraA01g035860.3C* (encoding WD40 protein), *BraA02g032320.3C* (encoding magnesium chelatase i2, *CHLI2*), *BraA05g025620.3C* (*ALWAYS EARLY 3*, *ALY3*), *BraA07g001310.3C* (encoding transducin family protein/WD40 repeat family protein, *FVE*), *BraA03g042340.3C* (encoding alanine: glyoxylate aminotransferase, *AGT*) and *BraA06g009540.3C* (encoding fumarylacetoacetase, *FAH*), which predictably interacted with *BrMYB28*, *BrMYB29*, *BrMYB34*, *BrMYB51* or *SUR1* and might be important candidate genes for Phe-derived GSL biosynthesis (Appendix A).

The candidate genes were screened around the 43 QTNs (100 Kb upstream and downstream region) for benzylGSL. Several known GSL biosynthesis genes were obtained, including *BraA07g038040.3C* (*MYB122*), *BraA07g038050.3C* (*SOT18*), *BraA07g038070.3C* (*SOT18*) and *BraA07g038080.3C* (*SOT16*). Two novel glucosyltransferases with different expression patterns in BrLG and BrHG, *BraA04g022840.3C* (*UGT74D1*) and *BraA04g022880.3C* (*UGT74C1*), were also located, and the homologous genes of *UGT74C1* were also regarded as candidate genes for GSL metabolism in *B. napus* [23]. Two candidate genes, *BraA01g035860.3C* (*WD40*) and *BraA06g009540.3C* (*FAH*), were predicted interacting with *BrMYB28*, *BrMYB29*, *BrMYB34*, *BrMYB51*, *BrMYB122* and *BrSUR1* (Appendix A). In addition, two novel candidate regulators for benzylGSL metabolism, *BraA02g038950.3C* (*MYB44*/*MYB77*) and *BraA08g032980.3C* (*BrMYB60*), were located around important QTNs with high contribution rate or repeatedly detected by different GWAS algorithms (Figure 2). These genes might be responsible for benzylGSL biosynthesis (Appendix A).

Similarly, the candidate genes were also identified around 59 QTNs for phenethylGSL. Two homologous genes to *MYB44*/*MYB77* (*BraA07g016630.3C*) and *MYB77* (*BraA03g044990.3C*), one cytochrome P450 family gene (*BraA03g061220.3C* (*CYP79B2*)) and one sulfotransferase gene, *BraA03g045180.3C*, were also detected. Four other candidate genes (*BraA02g032320.3C* (*CHLI2*), *BraA05g025620.3C* (*ALY3*), *BraA05g032450.3C* (*CYP72A9*) and *BraA07g001310.3C* (*FVE*)) were predicted to interact with known GSL biosynthesis genes (Appendix A). Furthermore, four other candidate genes were screened out, including *BraA02g037910.3C* (*MYB118*), *BraA03g061240.3C* (*AKN2*), *BraA06g024270.3C* (*GTR1*) and *BraA09g052850.3C* (*IGMT4*) (Appendix A), the homologous genes of which were also candidates for GSL metabolism in *B. napus* [23,24].

In addition, many known regulators and structural genes for GSL biosynthesis were detected on the hot regions of chromosomes A01, A02, A03 and A08, such as *BraA02g022090.3C* (*MYB122*), *BraA03g043830.3C* (*MYB34*), *BraA03g044380.3C* (*MYB28*), *BraA03g043740.3C* (*MAM1*), and *BraA08g028700.3C* (*MYB51*), etc., which might also have an effect on the accumulation of both of the Phe-derived GSLs (Appendix A).

### 2.4. Regulation of Phe-Derived Glucosinolates Biosynthesis Pathway in B. rapa Seeds

The expression levels of known and new-found candidate genes for Phe-derived GSLs biosynthesis were evaluated by FPKM value in developing seeds of BrHG and BrLG (Appendix A). The potential biosynthesis pathway for Phe-derived GSLs was constructed in *B. rapa* (Figure 3, Appendix A). The results showed that the candidate genes involved in Phe-derived GSL biosynthesis had different expression patterns between BrHG and BrLG (Figure 3, Appendix A), such as *CYP79C2*, *SOT16*, *MAM1*, *MYB28* and *MYB29*. In addition, the expression of two *CYP79A2* homologous genes (*BraA02g001640.3C*, *BraA10g028190.3C*) was not detected both in BrHG and BrLG, and several members of *CYP79C1*, *CYP79C2* and *CYP79F1* were differentially expressed between BrHG and BrLG, which also happened in some other genes (*SOT17*, *SOT18*, *MAM1*) (Figure 3, Appendix A). Several members of new-found candidate genes associated with benzylGSL (*BraA02g038950.3C* (*MYB44*/*MYB77*), *BraA08g032980.3C* (*MYB60*)) and phenethylGSL (*BraA02g037910.3C* (*MYB11*8)) had obviously up-regulated in BrHG, but *BraA03g044990.3C* (*MYB77*) and *BraA07g016630.3C* (*MYB44*/*MYB77*) displayed an opposite result. The results showed that gene members might have the functional redundancy and differentiation in *B. rapa*.

## 3. Discussion

As one of the most important secondary metabolites in plants, people attach importance to GSLs for the benefits for human health and plant defences or drawbacks for animal consumption. As one of the three kinds of GSLs in plants, benzenic GSLs have not received enough attention due to less species and content. In this study, we provide genomic and biochemical data for an increased understanding of two Phe-derived GSLs, including accumulation pattern and screening of previously known as well as novel and unique regulatory genes.

The dynamic accumulation pattern of the two Phe-derived GSLs, benzylGSL and phenethylGSL, was different in *B. rapa* seeds. According to the content of benzylGSL and phenethylGSL, phenethylGSL was the main Phe-derived GSL in *B. rapa* seeds. Although the content of benzylGSL in young seeds was higher in BrLG than in BrHG, which was lower in high mature (50 DAP) seeds of BrLG and might be part of the reason for the glucosinolate content difference between BrHG and BrLG. Although the content of two Phe-derived GSLs was low during 15 DAP, the structural genes catalyzing the synthesis of intermediate metabolites and benzenic GSLs were extreme highly expressed in this stage, which might be because the accumulation of metabolites lagged behind the expression of genes. However, another reason for the high expression of the structural genes could be involvement in biosynthesis of the remaining Met and Trp-derived GSLs in *B. rapa*.

Two different stereoisomers of hydroxylated phenethylGSL were dominant in *Barbarea vulgaris*, which were induced by *Plutella xylostella*, and the genes for chain elongation were all activated in different degrees [39]. In this study, *BCAT4* (*BraA05g027600.3C*), which catalyzes the first step of chain elongation, was with higher expression level in BrHG during 35 DAP (Figure 3), and the content of phenethylGSL also started to rise in BrHG in this stage (Figure 1). Except for *BCAT4*, other chain elongation genes were not significantly correlated with the content change of phenethylGSL (Figure 3), which might due to the existence of other long-chain GSLs in *B. rapa* (Appendix A). Same to the core structure biosynthesis genes.

The core structure biosynthesis genes for Phe-derived GSLs, Trp-derived indolic GSLs, Met-derived aliphatic GSLs, and the side-chain elongation genes for phenethylGSL and Met-derived GSLs are shared among different GSLs, the transcription factors regulating different GSLs biosynthesis were also shared [6]. It was verified that *MYB34*, *MYB51* and *MYB122* regulate indolic GSLs biosynthesis in *A. thaliana* and *B. oleracea* [17,40,41]. *MYB28* was verified controlling aliphatic GSLs biosynthesis in *B. juncea* and *B. napus* [14,23], which also potentially regulated benzylGSL biosynthesis in leaf and stem of *Sinapis alba* [25]. *MYB29* was responsible for aliphatic GSLs biosynthesis in *B. oleracea* [16,42], and also controlled root phenethylGSL variation in *B. napus* [28]. *MYB34* and *MYB29* were induced by *P. xylostella* and responsible for the biosynthesis of phenethylGSL in *B. vulgaris* [39]. In this study, these known genes were also obtained around linked QTNs on chromosome A02, A03 and A08 chromosomes (Appendix A), indicating that these genes’ members might also regulate the synthesis of Phe-derived GSLs in seeds of *B. rapa*. In addition, the contents of nine other GSLs (two indolic GSLs and seven aliphatic GSLs) were also calculated in BrLG and BrHG of *B. rapa*, the contents and accumulation dynamics were diverse from each other (Appendix A). How these regulators coordinate the metabolism of various GSLs needs further study.

Quantitative trait locus (QTL) mapping has been done on benzenic GSL content in *B. rapa* leaves, and three QTLs for phenethylGSL were located on chromosome A04 and A07 [7]. In this study, more loci were detected in other chromosomes except for A04 and A07, which can be attributed to the different organs used for measurement of GSL content and different methods used for mapping. Loci for GSLs were also identified in *B. napus* seeds and leaves by GWAS, and a serial of candidate genes were screened out for GSLs metabolism [43], 9 homologous genes of which were also obtained in our study, including *BrMYB34* (*BraA03g043830.3C*), *BrMYB28* (*BraA03g044380.3C*), *BrMYB118* (*BraA02g037910.3C*), *BrMAM1* (*BraA03g043730.3C*), *BrGTR1* (*BraA06g024270.3C*), *BrIGMT1* (*BraA09g052850.3C*), *BrSOT18* (*BraA07g038050.3C*, *BraA07g038070.3C*), *BrAKN2* (*BraA03g061240.3C*) and *BrUGT74C1* (*BraA04g022880.3C*). As the candidate genes for GSL biosynthesis in *B. napus* were the result of mapping on total GSLs’ content [43], these genes might be also involved in regulating the Phe-derived GSLs biosynthesis. Furthermore, new candidate genes for Phe-derived GSLs biosynthesis were found in this study based on the distinct QTNs for Phe-derived GSLs, transcriptome analysis and protein-protein interaction prediction, which needed further functional verification. Our research provided novel candidate genes for Phe-derived GSL accumulation in *B. rapa* seeds and broadened the understanding on different GSL metabolism.

## 4. Materials and Methods

### 4.1. Plant Materials and Sampling

One hundred fifty-nine *B. rapa* lines were collected from China and other countries (Appendix A), which were cultivated in Xining (Qinghai, China, N36°43′, E101°45′) in 2018 and 2020 from March to August and in Xiema (Chongqing, China, N29°76′, E106°38′) in 2019 from October to April. The leaf was sampled from one-month-old seedling of each line. The developing seeds of 35 days after pollination (DAP) were collected and mixed from 5 or more individuals for each line, furthermore, the developing seeds of 15, 25, 35, 45 and 50 DAP were also collected with two replicates in two *B. rapa* lines with high (BrHG) and low (BrLG) content of GSLs (the total content of GSLs), which were all quickly frozen in liquid nitrogen and were stored in a −80 °C ultra-low temperature refrigerator for long-term storage.

### 4.2. Glucosinolates Extraction

The raw metabolites were extracted from seed as previously described [44,45] with minor modifications. In brief, 0.1 g of fresh seeds stored at −80°C were quickly weighed and crushed into powder using high-throughput tissue grinder (Tissuelyser-192, Shanghai, China). Then 1 mL of extracting solution (80% aqueous methanol with 0.1% formic acid) was added and homogenized by vortex for 10 s. Then, the homogenized extraction buffer were treated with sonication (KQ-100E, Kunshan, China) at 4 °C for 1 h, followed by centrifugation at 8000 *g* at 4 °C for 30 min. In addition, the above process was repeated again on the precipitate. Finally, the two liquid supernatants were pooled and filtered with a 0.22-µm nylon filter for UPLC-HESI-MS/MS analysis. All experiments were performed at least three replicates for per accession.

### 4.3. UPLC-HESI-MS/MS Analysis

The filtered extract was performed on the Dionex UltiMate 3000 HPLC system (Thermo Fisher Scientific, Waltham, MA, USA) coupled to a Thermo Scientific Q-Exactive System equipped with an S-Lens ionizer source (Thermo Scientific, Waltham, MA, USA) as previously described [44,45] with minor modifications. The raw extractions were separated by an Acquity UPLC BEH C18 column (150 × 2.1 × 1.7 mm, Waters, Dublin, Ireland) with a guard column (1.7 μm particle size, 2.1 × 5 mm, Waters, Dublin, Ireland), thermostated at 40 °C. The mobile phase A is 0.1% formic acid (*v*/*v*), and B is 0.1% formic acid in acetonitrile (*v*/*v*). The elution program was 0−2 min, 5−10% solution B; 2−10 min, 10−25% solution B; 10−13 min, 25−95% solution B; 13−16 min, 95% solution B; 16−16.5 min, 95−5% solution B; and 16.5−21 min, 5% solution B. The flow rate was set to 0.300 mL/min, and the injection volume was 10 μL. The mass spectrometry was detected by the Full MS-ddMS^2^ method with negative model, which was ranged from 100 to 1500 (m/z). The gas of sheath, auxiliary and sweep were set to 35, 10, and 0, respectively. The source voltage was 3.5 kV with 350 °C capillary temperature.

### 4.4. Data Processing and Glucosinolate Identification

The UPLC−HESI−MS/MS data were analyzed using MS-DIAL 4.18 software with mass bank negative database (http://prime.psc.riken.jp/compms/msdial/main.html#MSP, accessed on 2 April 2022) [46], which were automatically converted by ABF (Analysis Base File) converter (http://www.reifycs.com/AbfConverter/index.html, accessed on 12 March 2020). In addition, the UPLC−HESI−MS/MS data were collected by Xcalibur 3.1 software and used to identify the compounds based on their characterized permanents, including retention times (RTs), accurate MS and MS/MS spectral data, together with the commercial standards and previously reported information [20,47,48,49]. Meanwhile, the unequivocally identified GSLs were quantified as our previously described, and sinigrin was used as a reference standard for drawing standard curve [44,45].

### 4.5. Transcriptome Analysis

The developing seeds of BrHG and BrLG during 15, 25 and 35 DAP were used for RNA extraction by EASYspin RNA Rapid Plant Kit (Bio-med, Beijing, China) with two replicates for each sample. A total of 12 qualified RNA samples were used for libraries construction by NEBNext^®^ UltraTM RNA Library Prep Lit for Illumina^®^ (NEB, Ipswich, MA, USA) and sequenced on Illumina Hiseq 2000 platform with 150 bp paired-end reads. After quality control, the clean reads were assembled and aligned to the *B. rapa* reference genome V3.0 (Chiifu) (http://39.100.233.196/#/Download/, accessed on 1 June 2021) [50] using Hisat2 [51]. Novel transcripts were predicted by StringTie [52], and transcript levels were calculated as FPKM (Fragments Per Kilobase of transcript sequence per Millions base pairs) with the featureCounts tool in Subread [53].

### 4.6. mGWAS Analysis of Phe-Derived Glucosinolates

Two Phe-derived GSLs, benzylGSL (C91) and phenethylGSL (C121), were identified and quantified in 35 DAP seeds of 159 *B. rapa* lines by UPLC−HESI−MS/MS. Meanwhile, DNA of 159 *B. rapa* local strains or inbred lines were extracted by Plant Genomic DNA Extraction kit (Tiangen, Beijing, China) and used for RAD-seq, and the reference genome was same as RNA-seq mentioned above. Fifty seven thousand five hundred fifty nine SNP markers evenly distributed in all chromosomes were extracted. STRUCTURE v2.3.4 was used for population structure analysis [54]. mrMLM v4.0 (https://cran.r-project.org/web/packages/mrMLM/index.html, accessed on 1 July 2021) was employed for GWAS using four algorithms, including mrMLM, FASTmrMLM, FASTmrEMMA and ISIS EM–BLASSO [55]. Genes located in 100 Kb downstream and upstream regions of linked quantitative trait nucleotides (QTNs) were obtained. Protein-protein interaction analysis was conducted using *B. rapa* as a reference in the STRING database (https://cn.string-db.org, accessed on 12 August 2021) [56]. The known genes participating in Phe-derived glucosinolate biosynthesis were collected in KEGG (Kyoto Encyclopedia of Genes and Genomes) pathway database (map00966: Glucosinolate biosynthesis, https://www.genome.jp/kegg/pathway.html, accessed on 17 August 2021). Combining mGWAS with the result of transcriptome analysis, candidate genes for Phe-derived glucosinolate metabolic were screened out.

## Figures and Tables

**Figure 1 plants-11-01274-f001:**
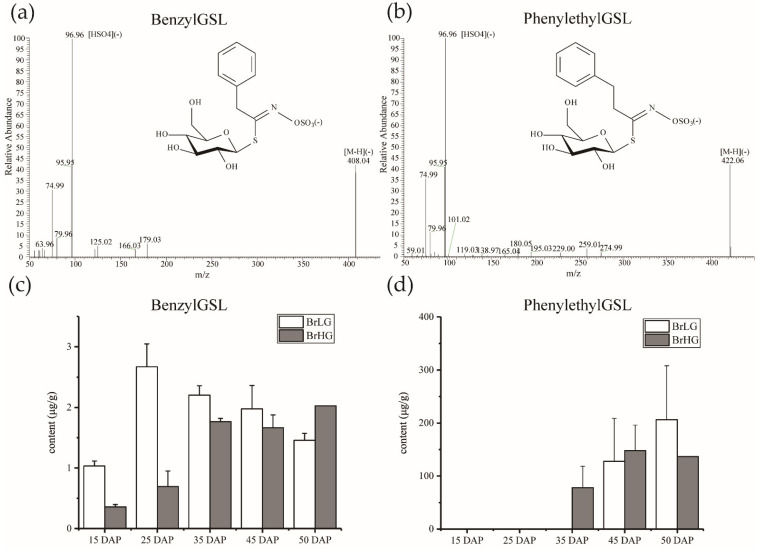
Characterization of Phe-derived GSLs in developing seeds of *B. rapa*. The fragmentation pattern (**a**,**b**) and dynamic accumulation (**c**,**d**) of benzylGSL and phenethylGSL in developing seeds of *B. rapa*. BrLG, *B. rapa* with low content of GSLs; BrHG, *B. rapa* with high content of GSLs. The content referred to the micrograms of GSL anion in per gram of fresh seed weight.

**Figure 2 plants-11-01274-f002:**
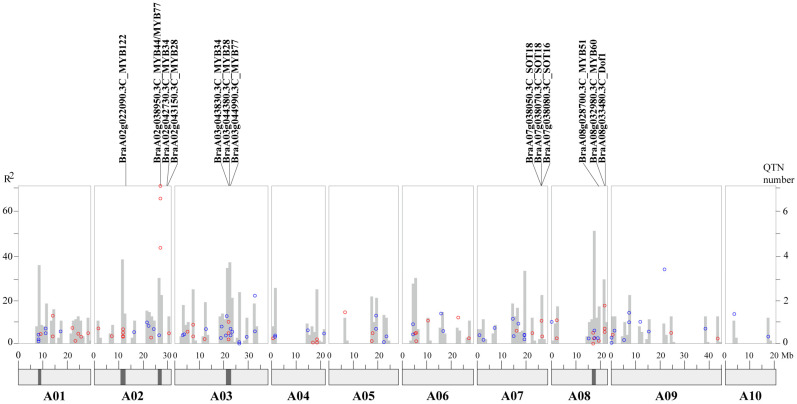
Distribution of QTNs for two Phe-derived GSLs by mGWAS in *B. rapa*. The red and blue dots indicate QTNs for benzylGSL and phenethylGSL, respectively. The gray bar indicates the numbers of detected QTNs on the chromosomes. The black of the chromosomes indicate five hot regions where QTNs gathered. R^2^ indicates the phenotypic variations explained by QTNs.

**Figure 3 plants-11-01274-f003:**
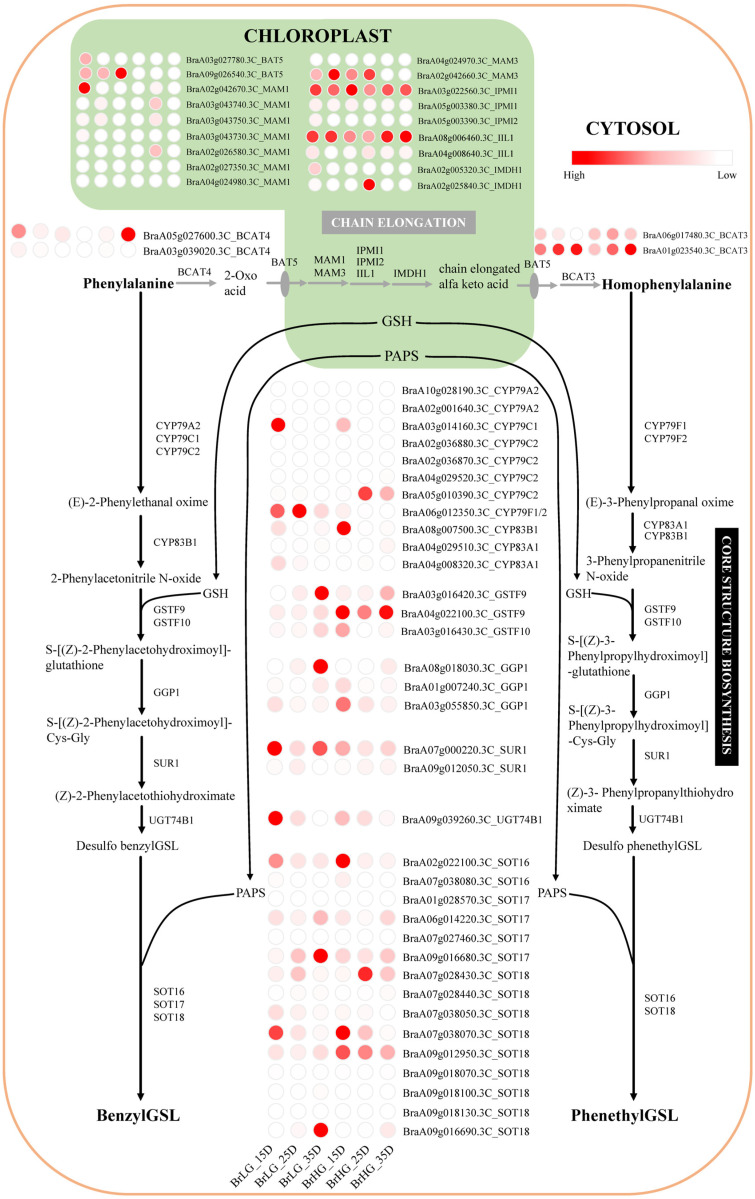
The proposed biosynthetic pathways of benzylGSL and phenethylGSL in *B. rapa*. The proposed pathways are based on the biochemical literatures [2,6,9,11,25]. The enzyme abbreviations refer to model plants and may need modification in the case of the *B. rapa* enzyme names. CYP, cytochrome p450 [11,12,13,26,27,28]; GSTF, glutathione S-transferase [29]; GGP, gamma-glutamyl peptidase [10]; SUR, S-alkyl-thiohydroximate lyase [30]; UGT, UDP-glucosyl transferase [31]; SOT, sulfotransferase [32,33]; BCAT, branched-chain aminotransferase [11,34]; BAT, bile acid transporter [35]; MAM, methylthioalkylmalate synthase [11,28]; IPMI1, isopropylmalate isomerase small subunit 1 [36,37]; IPMI2, isopropylmalate isomerase small subunit 2 [36,37]; IIL1, isopropylmalate isomerase large subunit 1 [37]; IMDH, isopropylmalate dehydratase [37,38]; BCAT3, branched-chain aminotransferase [34]; GSH, glutathione, γ-Glu-Cys-Gly; PAPS, 3′-phosphoadenosine-5′-phosphosulfate. The expression value of the structural genes in BrLG and BrHG was presented by heat map and more information was included in Appendix A.

## Data Availability

All other datasets supporting the results of this article are included within the article and its additional files.

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
