# Peer review of "Genome-Wide Association Study of Phenylalanine Derived Glucosinolates in Brassica rapa"

_plants, 2022, doi:10.3390/plants11091274_

Round 1
Reviewer 1 Report
see attached file

Author Response
Dear Reviewer and Editors,
Thanks for your comments concerning our manuscript entitled “Genome-Wide Association Study of Phenylalanine Derived Glucosinolates in Brassica rapa”. The comments have all been very helpful to us in revising and improving our paper, as well as being of value in guiding our research. We have studied the comments carefully and have modified the manuscript.
We also have revised all figures, tables and references according to the instruction of Authors of Plants, and reviewer’s comments in the revised MS. Revised portions are marked in the manuscript.
If you have any questions about it, please do not hesitate to tell us.
Thanks and best regards
Cunmin Qu

Reviewer 2 Report
The authors used metabolite based GWAS to uncover the genetics behind Phenylalanine-derived glucosinolates. Several loci were found to be important for GLS biosynthesis, some of which were previously known and some of which were novel. This study provides insight into GLS biosynthesis in Brassica rapa, and could be helpful for attempts to influence GLS accumulation for crop impovement. I found the methods good, the results were reported in a way that was easy to understand, and the writing was OK but could be improved.
Author Response
Dear Reviewer and Editors,
Thanks for your comments concerning our manuscript entitled “Genome-Wide Association Study of Phenylalanine Derived Glucosinolates in Brassica rapa”. We have studied the comments carefully and have modified the manuscript.
We also have revised all figures, tables and references according to the instruction of Authors of Plants, and reviewer’s comments in the revised MS. Revised portions are marked in the manuscript.
If you have any questions about it, please do not hesitate to tell us.
Thanks and best regards
Cunmin Qu

Round 2
Reviewer 1 Report
Dear authors, thank you for providing this information to the scientific community, I look forward to follow the use of this important paper over the coming years. I look forward to reading your future contributions to the glucosinolate community.
Minor edits plants benzenic GSL biosynthesis paper
Line 38: in order to get more citations I suggest some extra key words: benzylglucosinolate; phenethylglucosinolate; 2-phenylethylglucosinolate; glucosinolate biosynthesis and regulation
Line 43: Brassica genus (or Brassicaceae family or Brassiceae tribe), the present is botanically wrong
Line 75 remove “F”please correct to “….but also different CYP79 enzymes in the first step…”
Line 248 and 265 xylostella (missing a)
Line 61: add “a”: “…involves a single round…”.
Line 62: make the 2 lower case in CH2 according to usual chemical writing: CH2
Figure 3. one chemical name is still slightly wrong: the first intermediate in benzylGSL biosynthesis, just after phenylalanine, should be named (E)-2-Phenylethanal oxime. That is, “al” and a space should be added within the present name. In this way, you will see that the names in the two pathways correspond logically J.
Author Response
Dear Reviewer and Editors,
Thanks for your further comments for our manuscript. We have studied the comments carefully and have modified in the manuscript. Revised portions are marked in the manuscript.
Reviewer 1
Dear authors, thank you for providing this information to the scientific community, I look forward to follow the use of this important paper over the coming years. I look forward to reading your future contributions to the glucosinolate community.
Minor edits plants benzenic GSL biosynthesis paper
Line 38: in order to get more citations I suggest some extra key words: benzylglucosinolate; phenethylglucosinolate; 2-phenylethylglucosinolate; glucosinolate biosynthesis and regulation
Response: It was modified in the manuscript (Line 39-40).
Line 43: Brassica genus (or Brassicaceae family or Brassiceae tribe), the present is botanically wrong
Response: It was modified in the manuscript (Line 45).
Line 75 remove “F”please correct to “….but also different CYP79 enzymes in the first step…”
Response: It was modified in the manuscript (Line 78).
Line 248 and 265 xylostella (missing a)
Response: It was modified in the manuscript (Line 251 and 268).
Line 61: add “a”: “…involves a single round…”.
Response: It was modified in the manuscript (Line 64).
Line 62: make the 2 lower case in CH2 according to usual chemical writing: CH2
Response: It was modified in the manuscript (Line 64).
Figure 3. one chemical name is still slightly wrong: the first intermediate in benzylGSL biosynthesis, just after phenylalanine, should be named (E)-2-Phenylethanal oxime. That is, “al” and a space should be added within the present name. In this way, you will see that the names in the two pathways correspond logically J.
Response: Figure 3 was modified in the manuscript.
If you have any questions about it, please do not hesitate to tell us.
Thanks and best regards
Cunmin Qu
This manuscript is a resubmission of an earlier submission. The following is a list of the peer review reports and author responses from that submission.
Round 1
Reviewer 1 Report
please see the attached file
